# Changing Land Use and Urban Dynamics around an Industrial Zone in Bangladesh: A Remote Sensing Analysis

Palash Basak [1,2], Salim Momtaz [1,3,*], Troy F. Gaston [1] and Soma Dey [4]

1   School of Environmental and Life Sciences, University of Newcastle, Ourimbah, NSW 2258, Australia; palash.basak@northsouth.edu (P.B.); troy.gaston@newcastle.edu.au (T.F.G.)
2   Department of Environmental Science & Management, North South University, Dhaka 1229, Bangladesh
3   Centre for Sustainable Development, University of Liberal Arts, Bangladesh, Dhaka 1207, Bangladesh
4   Department of Women and Gender Studies, University of Dhaka, Dhaka 1000, Bangladesh; soma.dey@du.ac.bd
*   Correspondence: salim.momtaz@newcastle.edu.au; Tel.: +61-405678823

**Abstract:** This article examines the adverse effect of rapid industrialization around Bangladesh's Dhaka Export Processing Zone (DEPZ) by analyzing Landsat satellite images captured between 1989 and 2019. Image classification was performed to separate built-up areas with machine learning algorithms in Google Earth Engine. Image analysis was conducted using ArcMap and ArcGIS Pro. Field observations, interviews, and the literature review provided information for explanations about the phenomenon observed from satellite image analyses. The findings reveal that when DEPZ started its operation in 1993, there was hardly any built-up area in the vicinity. Within three decades, over 25% of the land within a 5 km radius of DEPZ has been converted into a built-up area, triggering an almost seven-fold increase in population. Industrial and urban growth in the DEPZ area has caused significant soil and water pollution in the broader region. As a result, the quantity and quality of agricultural land has degraded. In the long run, the planned industrial development initiative has contributed to unsustainable urban growth and environmental consequences. Insights drawn from this article can guide policymakers to re-evaluate their policy for rapid and large-scale industrialization.

**Keywords:** industrialization; urban growth; agricultural land; sustainable development; FDI; remote sensing; Google Earth Engine

## 1. Introduction

From a sustainable resource management and urban planning perspective, it is essential to understand the underlying cause of rapid change in land use and urban dynamics. Visual interpretation of satellite images shows that a vast area around the Dhaka Export Processing Zone (DEPZ) has converted into a built-up area, and the surface water quality in the region has changed quickly. A question arises whether the large-scale industrialization in this area is associated with accelerated urban growth and environmental degradation.

Industrial growth in Bangladesh is primarily driven by the ever-increasing demand for outsourcing from the developed part of the world [1]. Global companies are making Foreign Direct Investment (FDI) and setting up their industrial production units by taking the benefits of cheap labor, relaxed environmental regulations, and the overall low product cost this developing country offers [2]. DEPZ is one of the oldest among nine Export Processing Zones (EPZs) in the country and has attracted a significant flow of FDI since the 1980s. While DEPZ earns billions of dollars in foreign currencies, offers primary and secondary job opportunities, and helps the economy grow faster, does it significantly influence secondary industrial growth, population redistribution, and environmental pollution?

Studies unfold the positive connection between FDI flow from developed countries and environmental degradation in developing countries [3,4]. The Pollution Haven Hypothesis

(PHH) suggests that investors from developed countries usually shift dirty and labor-intensive industries to developing countries [5]. Foreign investors are taking advantage of developing countries' relaxed environmental standards and regulatory framework, which helps them reduce overall production costs [6,7]. However, some researchers [8,9] have denied the findings of those studies that supported PHH and suggested that FDIs instead support developing economies.

While there is debate about whether developed countries are exporting their polluting industries or not, it is undeniable that developing countries are experiencing significant environmental degradation because of speedy industrialization in recent decades [8]. Much research is devoted to analyzing land use change [9], industrial growth [10], and environmental degradation in developing countries [10–12]. However, no significant study has been found that investigates the linkage between FDI and urban growth based on direct observational data, such as satellite images. It was observed that there is a lack of research on the long-term and cumulative effects of FDI-induced industrialization from a sustainability perspective. This study, therefore, has been designed to investigate the association between FDI and unsustainable urban growth using DEPZ as a case study and by examining archived satellite images as empirical evidence.

This study has used Google Earth Engine (GEE), a cloud-based geospatial data and analytical solution from tech giant Google, as it has a massive archive of satellite images (https://earthengine.google.com/ (accessed on 10 July 2023)). GEE offers faster and more efficient analytical abilities. With this cloud-based application, Google envisioned doing the heavy-lifting part of data storage, retrieval, and processing to the cloud so that scientists can perform analysis of geospatial data to find solutions for climate change and other global environmental issues. Google published several tutorials as well as documentation of the application. One of the preliminary and essential articles on the functionality, vision, and potential use cases of GEE was published by Gorelick et al. [13]. Subsequently, a good number of papers were published to demonstrate the functionality of GEE in the spatio-temporal analysis of environmental phenomena, including changing patterns of forest [14], agriculture [15,16], water, land use [17,18], temperature, air quality and so on.

However, most studies conducted with GEE have a relatively larger geographical scale. Few papers could be found where local-level analysis was performed with GEE. This study was an attempt to do so. The application of the machine learning algorithm for this study is unique in that it used known urban footprints from recent years and classified older images. Most studies that used machine learning algorithms generally classified images based on samples from the same time frame [17,19,20]. The classification method used in this study eliminates the need for collecting training samples from different time frames and allows a more straightforward classification of historical images.

It is hypothesized in this research that the accelerated growth of the built-up area in the vicinity of DEPZ occurred because of the establishment of an FDI-induced industrial zone. Built-up area refers to both industrial and residential development. In this connection, this study aims to assess the pattern of urban footprint expansion around DEPZ using satellite images available in GEE. The four objectives of this study are (a) to examine the growth pattern of built-up area in the last 30 years (1989 to 2019) around DEPZ by using historical Landsat satellite images; (b) to investigate the association between the built-up area expansion and population growth; (c) to explore the connection between the industrialization and pollution; and (d) to assess whether GEE is an effective tool for small scale studies like this one.

## 2. Materials and Methods

Satellite image classification and analysis are this research's primary methods of investigation. However, additional data and information from different sources contextualize and validate the findings from image analysis. The following diagram (Figure 1) shows the steps of the research method. Subsequent subsections discuss the study area and the research method in more detail.

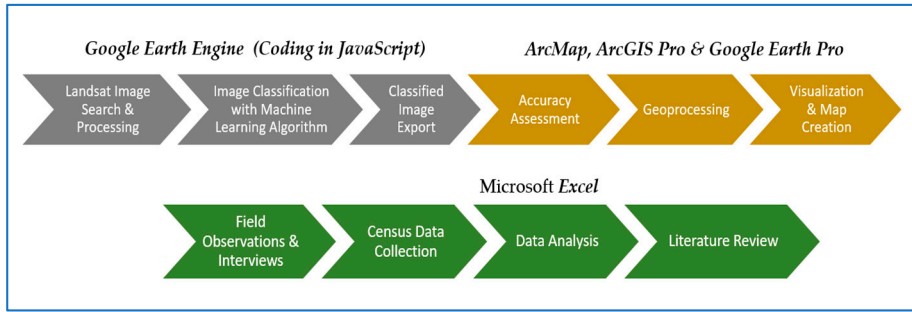

**Figure 1.** Key steps of the study.

*2.1. Study Area*

The study area covers approximately 80 square kilometers around DEPZ (Figures 2 and 3). It is located about 20 kilometers (km) northwest of Dhaka CBD. Around 100 industries in the industrial zone have attracted approximately US $1.2 billion in FDI in the last two decades (BEPZA, 2016). DEPZ currently contributes over US $2 billion yearly to the Bangladeshi economy, employing over 90 thousand workers (BEPZA, 2016). Enterprises from 30 countries, including South Korea, China, UK, USA, Germany, and Japan, have invested in the industrial zone. Over half of the industries in DEPZ produce ready-made garments or garment accessories. Another one-third of the manufacturing units are related to textile, knitting, and textile-related products. The rest manufacture plastic materials, footwear, leather goods, etc. (http://bepza.gov.bd (accessed on 25 July 2023)).

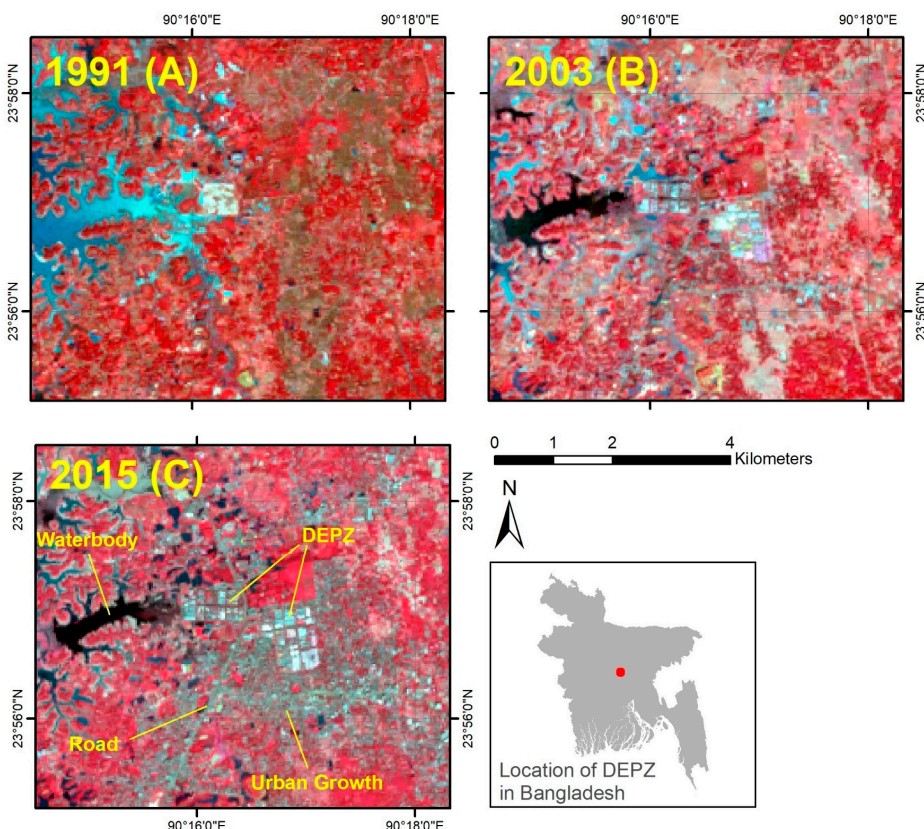

**Figure 2.** False color composition (NIR, red and green band) of Landsat Image for 1991 (**A**), 2003 (**B**), and 2015 (**C**) showing Dhaka Export Processing Zone (DEPZ) and its surrounding areas. Urban growth and changes in watercolor are visible in the images.

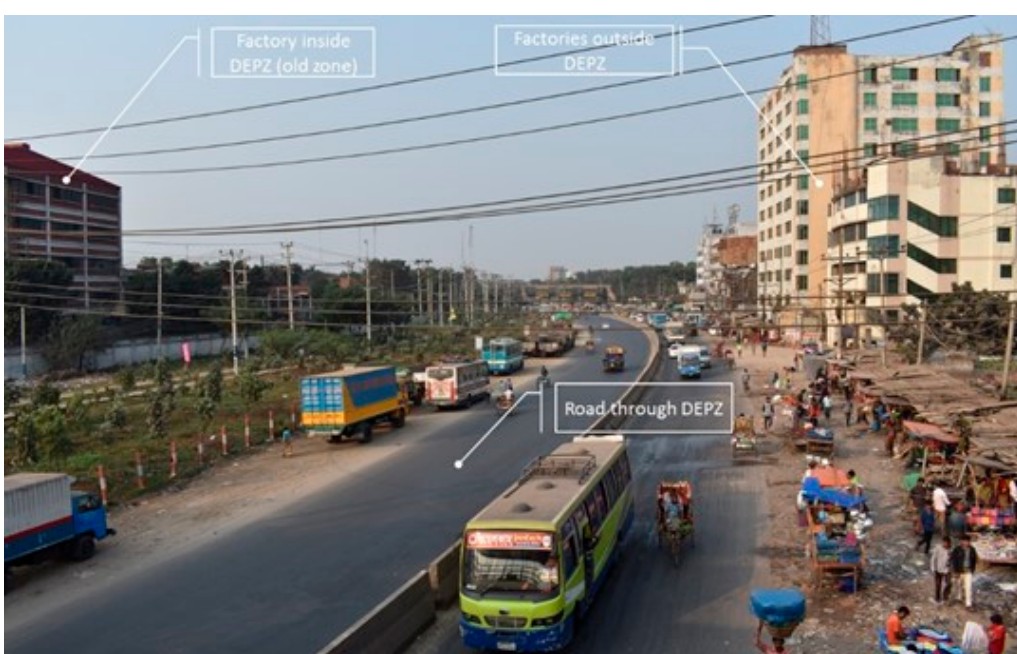

**Figure 3.** Photograph showing the main road running through the DEPZ area, view from the foot-over bridge near the main gate of DEPZ's new zone. Factories outside the DEPZ boundary are visible in the image. The photo was taken during fieldwork in June 2018.

### 2.2. Satellite Image Dataset and Processing Steps

Two GEE datasets, namely USGS Landsat 8 Collection 1 Tier 1 TOA Reflectance (LANDSAT/LC08/C01/T1_TOA) and USGS Landsat 5 TM Collection 1 Tier 1 TOA Reflectance (LANDSAT/LT05/C01/T1_TOA) were used in the analysis. Landsat data were used in this study because this is the most prolonged and consistent satellite image program, with global coverage and an open data access policy. A JavaScript-based code was developed for this research to perform all the analyses in GEE. The code can be accessed from the web link provided in the Supplemental Material section. All analyses performed in GEE for this study could be reproduced using the code.

The first step of the analysis in GEE was to filter Landsat 8 and Landsat 5 image collections to obtain images for the study area. A total of seven Landsat satellite images—representing every five years from 1989 to 2019—were used in this study (Table 1). The images were filtered out for seasonality and cloud coverage. The least cloudy images—acquired during January or February of each representative year—were chosen. The cloud coverage of the selected images was between zero (0) and six (6) percent (Table 1).

**Table 1.** Details of the Landsat images and trained classification model used in the study.

| Sensor/ Representative Year | Image Acquisition Date | Cloud Coverage (%) | Used for Training | Overall Accuracy of Trained Classification Model (%) | Validated Classified Image | Validated Accuracy of the Classified Image (Kappa Statistics) |
|---|---|---|---|---|---|---|
| Landsat 8 | | | | | | |
| 2019 | 23 January 2019 | 0 | Yes | 95.6 | Yes | 0.92 |
| 2014 | 26 February 2014 | 2 | | | Yes | 0.88 |
| Landsat 5 | | | | | | |
| 2009 | 11 January 2009 | 1 | Yes | 98.9 | Yes | 0.94 |
| 2004 | 15 February 2004 | 6 | | | Yes | 0.90 |
| 1999 | 1 February 1999 | 0 | | | | |
| 1994 | 2 January 1994 | 0 | | | | |
| 1989 | 4 January 1989 | 3 | | | | |

Six Landsat 8 bands (Band 2 to Band 7) and seven Landsat 5 bands (Band 1 to Band 7) were used for image classification. In addition to visible and infrared spectral bands, three indices, i.e., the Normalized Difference Vegetation Index (NDVI), Normalized Difference Built-up Index (NDBI), and Modified Normalized Difference Water Index (MDNWI) were calculated and included in the classification process. The following equations were used to calculate the indices from each Landsat image.

$$NDVI = \frac{(NIR - RED)}{(NIR + RED)} \tag{1}$$

$$NDBI = \frac{(SWIR - NIR)}{(SWIR + NIR)} \tag{2}$$

$$MNDWI = \frac{(GREEN - SWIR)}{(GREEN - SWIR)} \tag{3}$$

where NIR = near-infrared range, RED = red range, SWIR = short-wave near-infrared range, and GREEN = green range of reflectance in the spectrum [21–23].

### 2.3. Image Classification, Analysis and Visualization Process

Two different classifiers were constructed—one for Landsat 8 and one for Landsat 5 images. Both classifiers were based on Support Vector Machine (SVM)—a machine learning algorithm associated with a non-parametric supervised learning model. SVM maximizes class separation by projecting training data into multidimensional space and finding hyperplanes [24]. Three custom parameters of SVM, i.e., kernel type (RBF-Radial Basis Function), gamma (0.5), and cost (10), were applied. The classifiers were found to be effective in identifying built-up areas. Internal consistency or accuracy of the classifiers was over 95% (Table 1). The base image for the Landsat 8 classifier was taken from 2019, while that for Landsat 5 was taken from 2011. Older images were classified using these classifiers. Training samples (for built-up and non-built-up areas) for the classifiers were created in Google Earth Pro on Desktop (https://www.google.com/earth/versions/#earth-pro (accessed on 12 March 2023)) using historical high-resolution images as a backdrop.

The built-up area of 2019 was used as a mask for older classified images. The idea behind this strategy is that the older built-up area should be detected in the new image. This strategy helped filter misclassified images beyond the known urban boundary identified from the latest images. Landsat-8 classifier was used to obtain the built-up area of 2019 and 2014. Landsat-5 classifier was used for obtaining built-up areas for 2009, 2004, 1999, 1994, and 1989.

The classified images were further validated with the post-classification accuracy assessment process in ArcGIS Pro (version 2.4, https://www.esri.com/en-us/arcgis/products/arcgis-pro/overview (accessed on 15 June 2023)). The validation was performed based on higher-resolution images available in Google Earth Pro. In the context of this study, historical high-resolution satellite images available in Google Earth Pro were utilized as a proxy for field observation data. Because higher-resolution images before 2001 were unavailable in Google Earth for the study area, validation with high-resolution images was restricted to 2004. At least thirty random points were generated to assess the validity of each classified image. Based on the analysis, the classified images have a higher level of accuracy (Kappa statistics $\geq$ 0.88, Table 1).

A composite image was constructed by adding classified images to find newer and older urban growth. Outputs from GEE were exported to Google Drive (https://www.google.com/drive/ (accessed on 12 June 2023)) and then to personal computers. ArcMap (version 10.6.2, http://desktop.arcgis.com/en/arcmap/ (accessed on 15 June 2023)) and ArcGIS Pro were used for visualization, validation, and subsequent analysis of exported images from GEE. The association between the growth of urban built-up areas and population growth in the study area was explored with correlation and regression analysis using Microsoft Excel.

*2.4. Field Observations and Literature Review*

As this study was conducted as part of Ph.D. research, extensive fieldwork was conducted around DEPZ by the first author between 2016 and 2018. Besides making field observations and taking geo-tagged photographs, the first author conducted 15 semi-structured interviews with subject-matter experts and local people. The first and second authors have been familiar with the study area even before the establishment of DEPZ. The secondary literature from journals and newspapers (one English daily, The Daily Star (https://www.thedailystar.net/ (accessed on 28 July 2023)), and one Bangla daily, Prothom Alo (https://www.prothomalo.com/ (accessed on 30 July 2023)) were reviewed. Population data had been collected from the Statistical Yearbooks from the Bangladesh Bureau of Statistics (BBS) library located in Dhaka. All these activities helped relate the results of satellite image analysis with FDI, urban growth, and population expansion.

**3. Results**

The spatial extent of the built-up area for every representative year between 1989 and 2019 has been presented in Figure 4, which shows a sharp increase in built-up area in the latter years. Before the establishment of DEPZ in 1993, there was almost no built-up area near DEPZ (about 1%). One year after DEPZ started operating, the extent of the built-up area was still low. In 1999, the built-up area within a 5 km radius of DEPZ became 2.5 sq. km, translating to approximately 3% of the total area. The built-up area continued to grow in the following years; it became around 7% and 20% in 2004 and 2014, respectively. By 2019, approximately 28% of land within a 5 km radius of DEPZ was converted to a built-up area (Table 2). High-resolution satellite images also depict the recent changes in the area (Figure 5).

The speed of urban growth in the study area increased over time. For instance, the development of urban built-up areas in the first five years after the establishment of DEPZ (between 1994 and 1999) was only 2% (Table 2). However, the five-year growth of the built-up area became 6.8% between 2009 and 2014, and 8.0% between 2014 and 2019 (Table 2). A second-order polynomial equation could explain the growth of the urban built-up area around DEPZ (Figure 6). The non-linearity of the 5-year urban growth trend line indicates that faster urban expansion has happened in recent years.

The composite version of the classified image indicates that the older built-up area was concentrated near the DEPZ and primary roads (Figure 7). On the other hand, new growth in built-up areas occurred further away from the major road network. Figure 7 shows that the southeast part of DEPZ went through the highest level of development in terms of built-up area. The most negligible growth in the built-up area happened in the western, northeastern, and southwestern parts of DEPZ. This pattern of built-up area growth can be explained by the fact that there is a large water body in the west, a pre-existing government establishment (Atomic Energy Research Establishment) in the northeast, and a cantonment (Savar Cantonment) in the southwest. The southeast part of DEPZ had relatively elevated land.

**Table 2.** Growth of built-up area * within a 5 km radius of DEPZ.

| Year | Built-Up Area (Sq. km) | Built-Up Area (%) | Growth of Built-Up Area in Previous Five Years (%) |
|------|------------------------|-------------------|----------------------------------------------------|
| 1989 | 0.85 | 1.1 | - |
| 1994 | 0.90 | 1.1 | 0.1 |
| 1999 | 2.50 | 3.2 | 2.0 |
| 2004 | 5.31 | 6.8 | 3.6 |
| 2009 | 10.67 | 13.6 | 6.8 |
| 2014 | 15.98 | 20.3 | 6.8 |
| 2019 | 22.24 | 28.3 | 8.0 |

* Estimated based on classified images of this study.

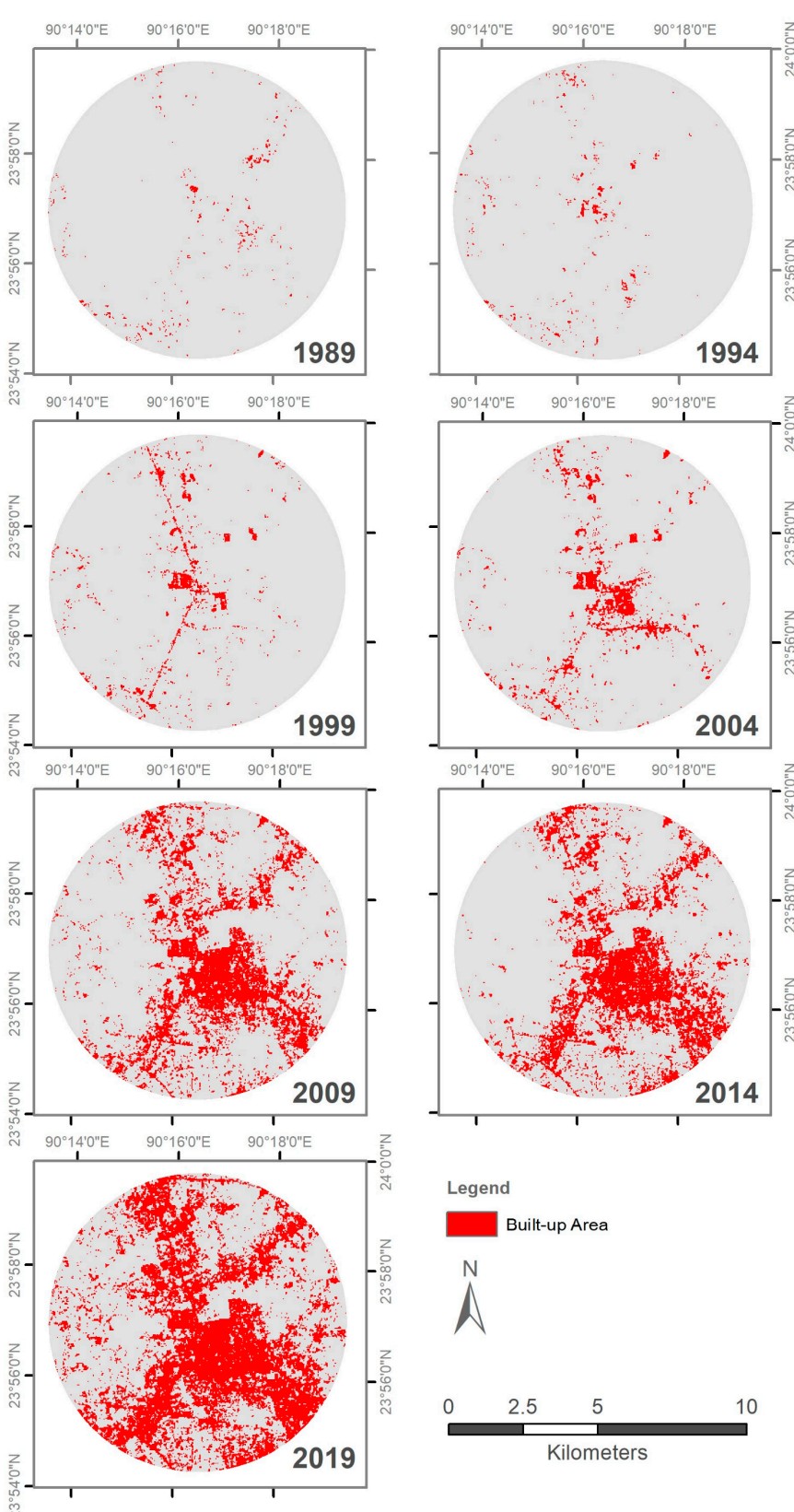

**Figure 4.** Growth of built-up area in the last 30 years within a 5 km radius of DEPZ–1989–2019 (one image per representative year). The sequence of images shows that the speed of urban growth accelerated over the years.

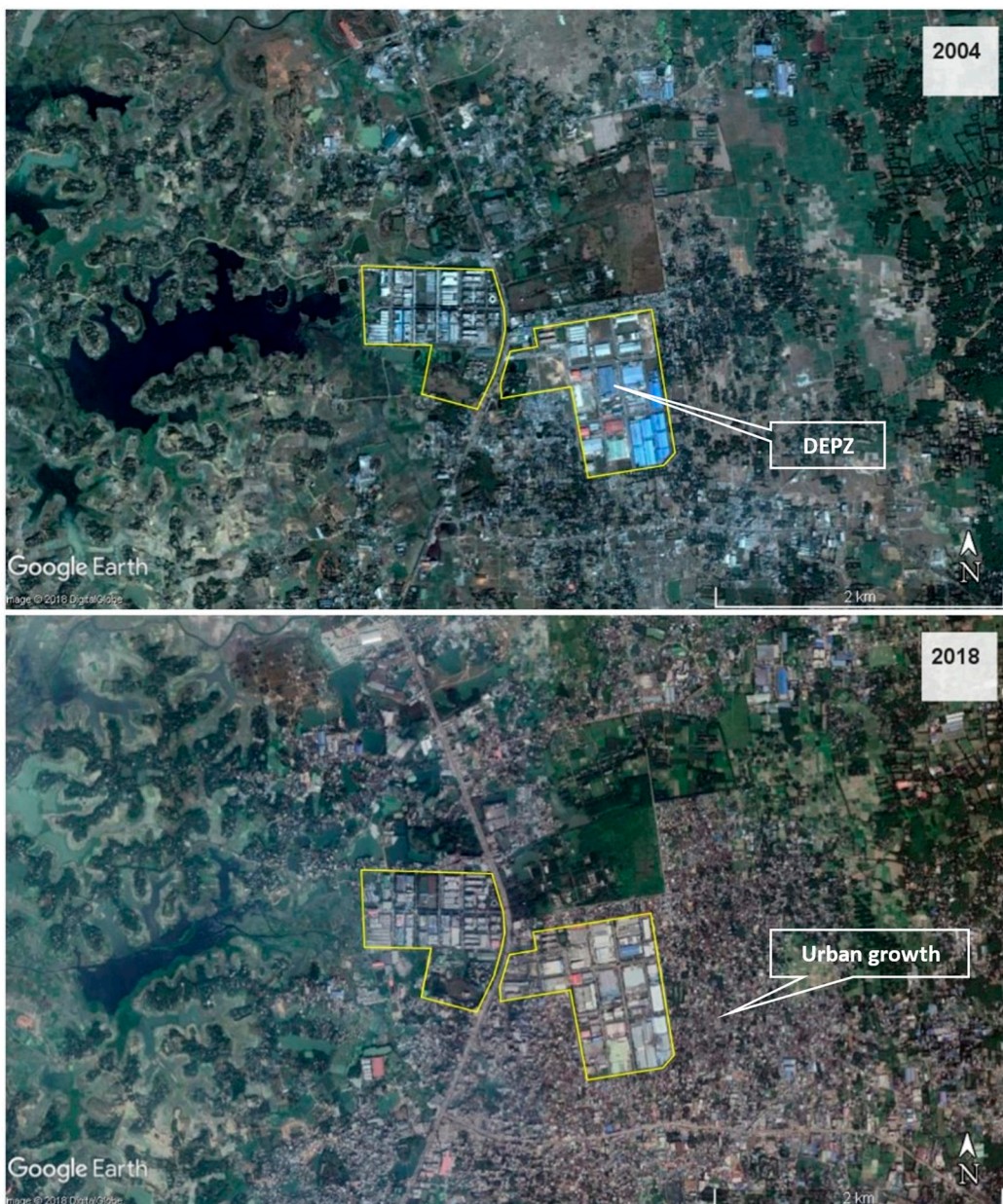

**Figure 5.** Google Earth images showing DEPZ and surrounding areas in 2004 and 2018. Compared to 2004, significant urban growth around DEPZ is visible in the 2018 image.

Like built-up areas, the population around DEPZ has grown dramatically. Figure 8 indicates that all unions (lowest local government tier in Bangladesh) around DEPZ experienced significant growth in population. Dhamsona Union, where DEPZ is located, experienced the highest population growth. In the 2011 census of Bangladesh, the union was identified as the largest union in the country in terms of population [25]. At that time, the union had over 300,000 people with a density of approximately 10,000 per square kilometer. The current population of the union is estimated to be about 500,000 or more, whereas the 1991 population was only about 33,000.

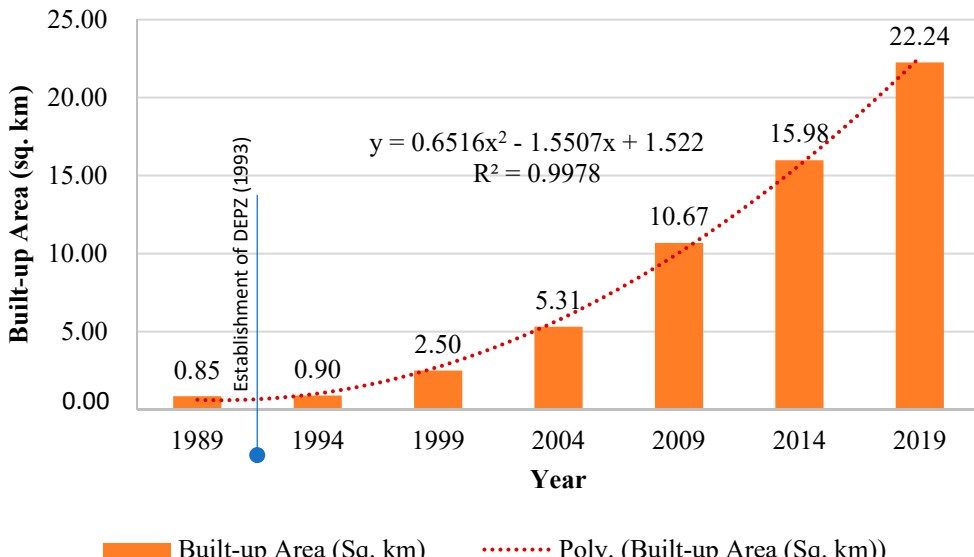

**Figure 6.** Growth of built-up area within a 5 km radius of DEPZ between 1989 and 2019.

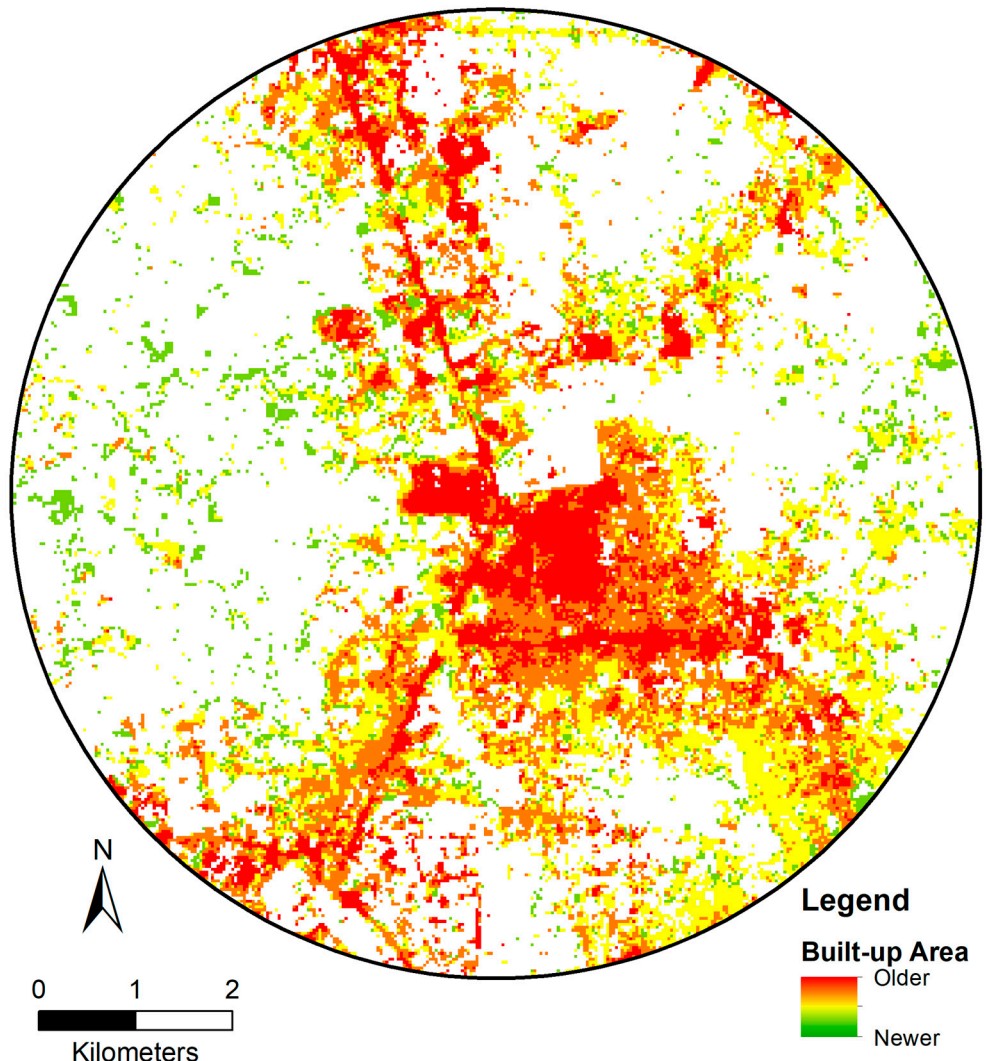

**Figure 7.** Growth of built-up area in the last 30 years within a 5 km radius of DEPZ–1989–2019 (all classified images merged).

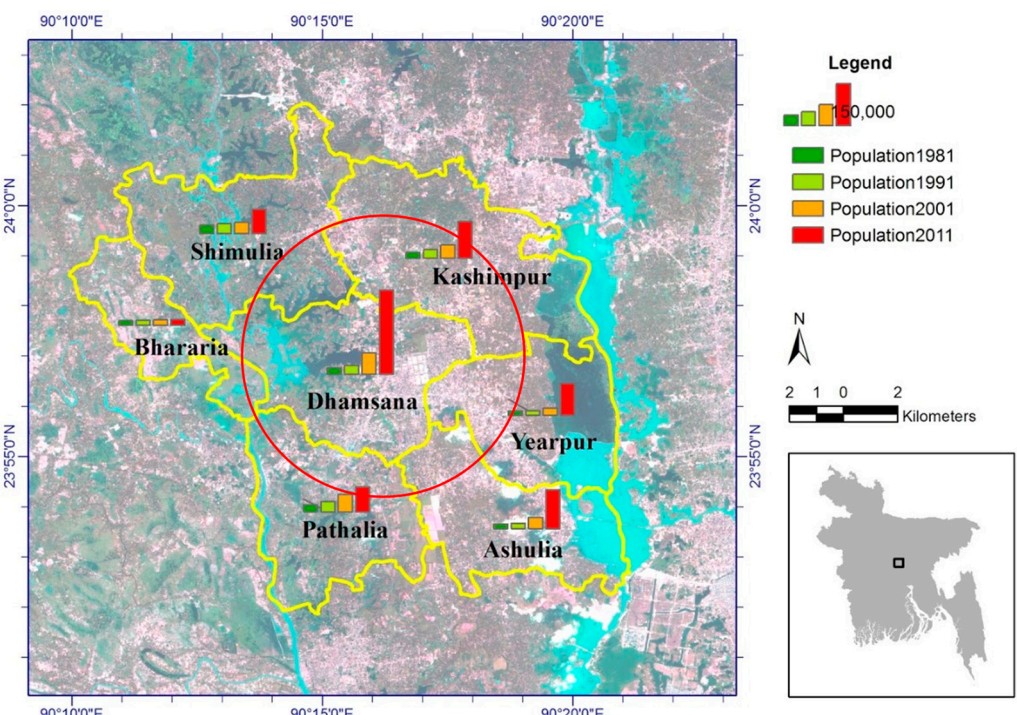

**Figure 8.** Census data indicating demographic change in unions around the DEPZ area. Dhamsona Union, where DEPZ is located, experienced the highest population growth. The red circle indicates 5-km buffer around DEPZ and the black box inside the inset map indicates the main map area in Bangladesh.

The amount of built-up area and population after two years in the nearby unions of DEPZ shows a very strong positive linear association, $r$ (5) = 0.99, $p$ < 0.001 (Table 3 and Figure 9). Even though there are limited data on population, it can be inferred based on the linearity between the two variables that when the built-up area increased, the population also increased. The pace of population growth was proportional to the development of the built-up area. Theoretically, every square kilometer increase in the built-up area within a 5 km radius of DEPZ resulted in a significant change in the local population structure; around 70 thousand additional population had been added in the seven unions around DEPZ (Figure 9).

The median Landsat satellite images revealed that the quantity and quality of surface water around the DEPZ were altered over time. Before DEPZ was established, between 1988 and 1992, an average 5242 square meters area within a 5-km radius of the industrial center was covered with surface water (estimated based on MNDWI). The spatial distribution of the median images for the three periods confirms the reduction in the volume of the surface waterbody.

**Table 3.** Built-up area within a 5 km radius of DEPZ and total population in seven unions * around DEPZ.

| Year | Built-Up Area (Sq. km) | Total Population in 7 Unions * around DEPZ (with a Two-Year Lag) |
|---|---|---|
| 1989 | 0.85 | 197,000 |
| 1994 | 0.90 | 270,000 ** |
| 1999 | 2.50 | 319,000 |
| 2004 | 5.31 | 673,000 ** |
| 2009 | 10.67 | 909,000 |
| 2014 | 15.98 | 1,371,000 ** |
| 2019 | 22.24 | 1,679,000 ** |

* The unions are Dhamsona, Ashulia, Kashimpur, Yearpur, Pathalia, Shimulia, and Bhararia. ** Estimated.

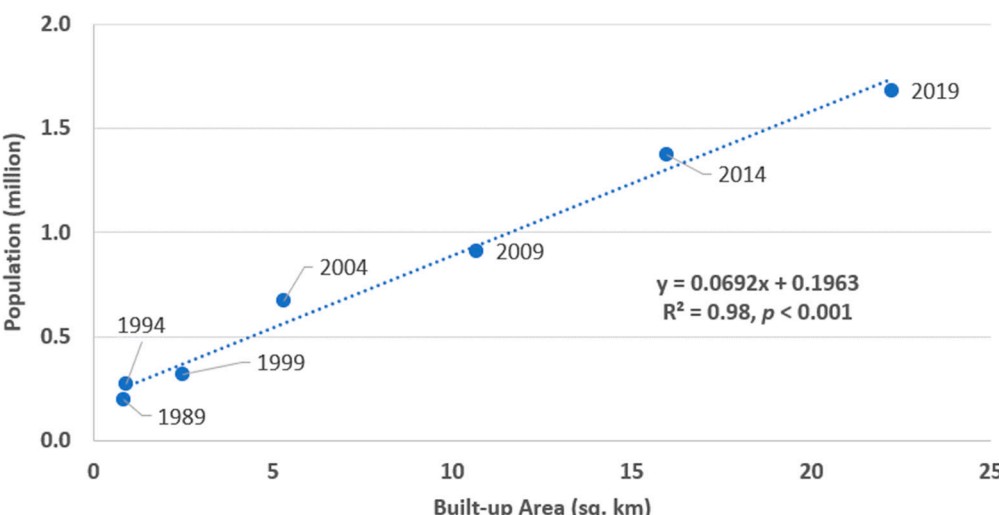

**Figure 9.** Built-up area vs. population after two years in unions around DEPZ. The two variables show a significant positive association.

Ironically, the localities around DEPZ are still considered rural areas even though these settlements have over one million people, and the population density has been higher than many municipalities in the country. Till today, the study area is administered by a rural local government system. No municipality has been set up. Union-level local government in Bangladesh does not have any technical arm for planning. The urban growth around DEPZ could, therefore, be tagged as unplanned.

The median reflectance of Landsat images captured between 1988 and 1992 indicates (Figure 10) that Dhalai Beel was initially a seasonal water body. That is why the waterbody was hardly identifiable from the pre-DEPZ median image. However, the border of Dhalai Beel is evident from the median reflectance of Landsat images captured between 1994 and 2011. At the same time, the extent of the Beel also increased. That means Dhalai Beel became a permanent waterbody afterward. It was also observed that the median reflectance between 1994 and 2011 appears darker. Similar conditions remain for the period between 2013 and 2019. During this time, the median extent of the Beel has been reduced significantly. However, darker reflectance persists. A comparison of the median reflectance shows that Dhalai Beel's water has become darker than before the DEPZ period.

During the field visit, it was observed that the water near the effluent discharge point of DEPZ was almost black (Figure 11). Heavy smells were coming out from the water. Yet, kids and grown-ups were fishing in that water. The fish were hardly edible. A large fishpond, directly connected to the filthy water of Dhalai Beel, was observable in the area. Fishing boats and fishing nets were seen in the Beel. During the monsoon period, flood water becomes mixed with the Dhalai Beel water. Floods are typical in the area—they happen nearly every year. As a result, the industrial effluent of the DEPZ area becomes mixed up with the nearby Dhaleshwari/Bonshi river water and is carried over downstream. Field observations reveal that the Dhalai Beel and almost all surface water bodies around DEPZ are filthy and polluted. Factories outside the DEPZ appeared to be more polluting than the inside ones because outside factories rarely have an effluent treatment plant. According to a local journalist, there are at least four times more factories outside than inside DEPZ.

Because of polluted water, agricultural production in a vast area has been stopped. At the same time, contaminated water is used for irrigation and fish cultivation. Contaminants from the industries are likely moving to the food stream. Researchers found the presence of trace metals in soil and rice [26,27].

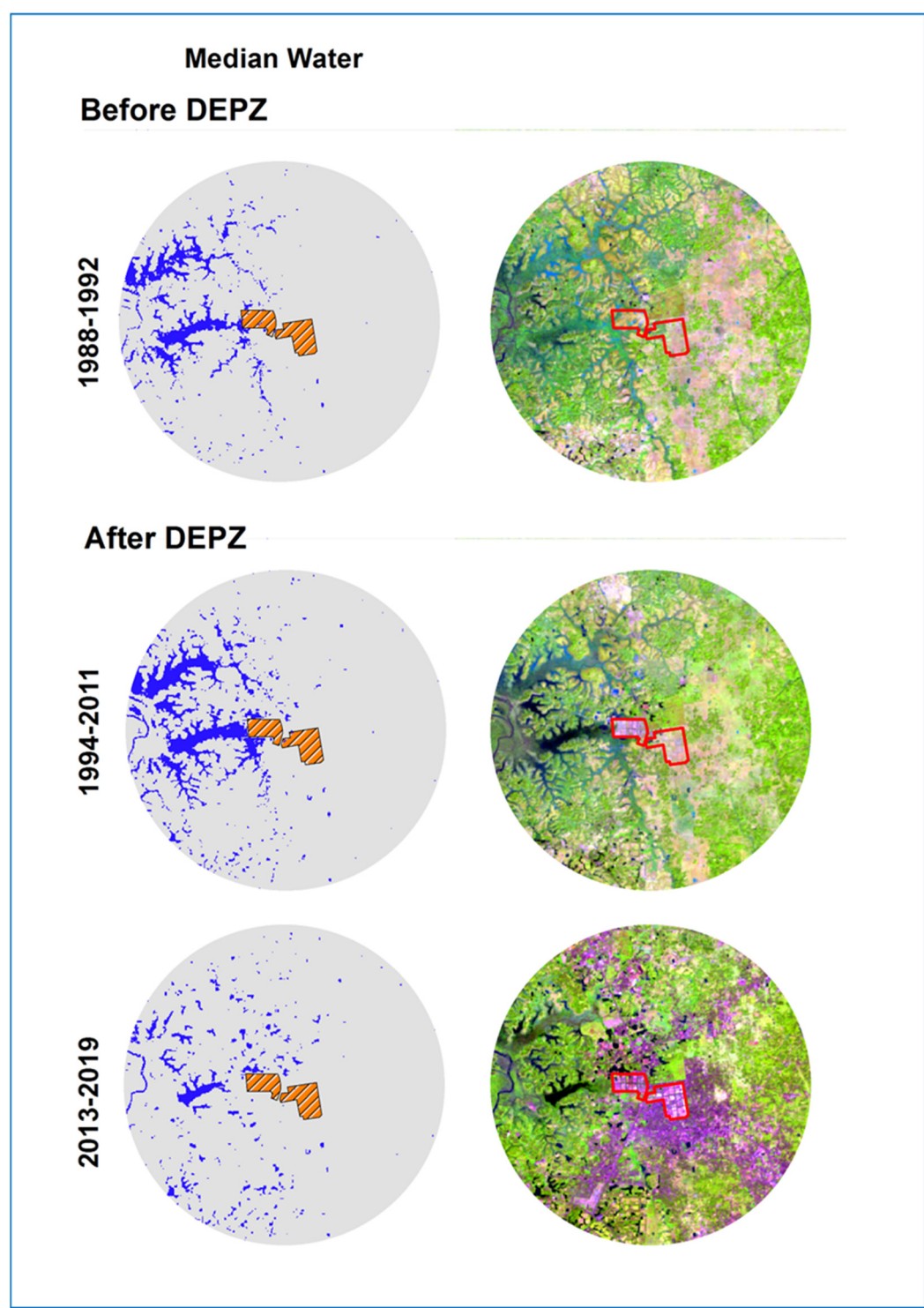

**Figure 10.** Change in water quantity and quality around DEPZ depicted by water index (MNDWI) and false color composition of median Landsat reflectance.

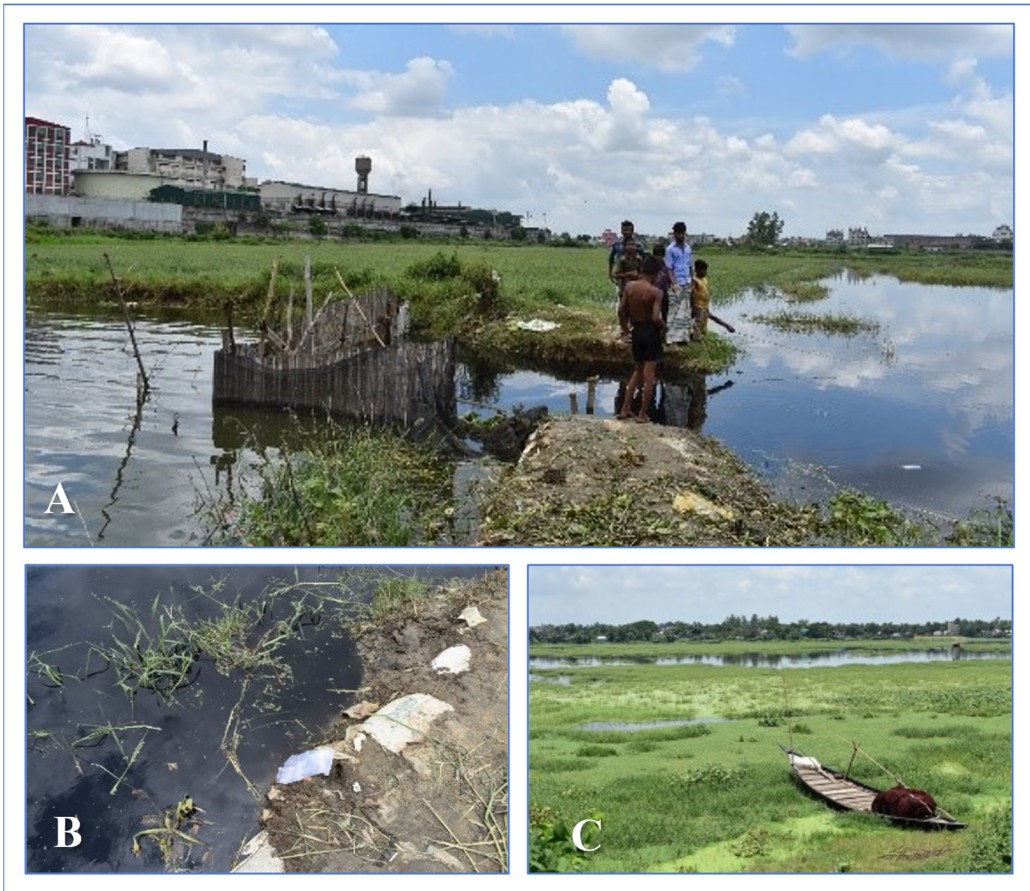

**Figure 11.** Photos show the surface waterbody (Dhalai Beel) near the discharge point of the DEPZ. (**A**) People are farming and catching fish in the contaminated water. (**B**) Blackish and smelly water. (**C**) Fishing boat and net. Photos were taken during fieldwork in June 2018.

## 4. Discussion

Bangladesh, an emerging economy in South Asia, has been experiencing accelerated urban growth in the last few decades [28]. More than one-third of the population of this country now lives in urban areas [29,30]. As the capital city of Bangladesh, Dhaka is experiencing the highest level of urban growth in the country and the world [31]. The peripheral areas of Dhaka are experiencing change at a higher pace [28]. The rapid urban growth around Dhaka can be explained by establishing industrial units, especially by expanding ready-made garment (RMG) factories [32]. RMG factories require a lot of manual labor and attract an unskilled workforce (primarily women), who generally migrate to the cities from rural areas [33,34]. According to a recent study, over 70% of RMG factories in Bangladesh are situated in the greater Dhaka region [35]. RMG factories are the majority in the DEPZ Area.

Findings presented in the previous section indicate that the built-up area and population around DEPZ have grown faster in the last few decades. DEPZ has been the center of all development in the area. However, the accelerated growth of built-up areas and population can be considered a result of a series of events. First, the industries in the FDI-induced EPZ create employment opportunities, attracting local and outside laborers. The increased population needs services that create opportunities for additional employment and business. Facilities such as markets, schools, hospitals, mosques, cinema halls, etc., appear (Figure 12). Second, factories outside DEPZ began to pop up by taking advantage of the upgraded infrastructures, labor availability, and weak administration (Figure 13). Third, further urban expansion happens in the surrounding areas of DEPZ as a combined result of industrial and residential development. All these events have occurred in a loop.

The current urban footprint and population explosion around DEPZ are the cumulative result of that loop. It could, therefore, be concluded that the FDI-induced industrial development is directly linked with the subsequent growth in the built-up area and population in its vicinity.

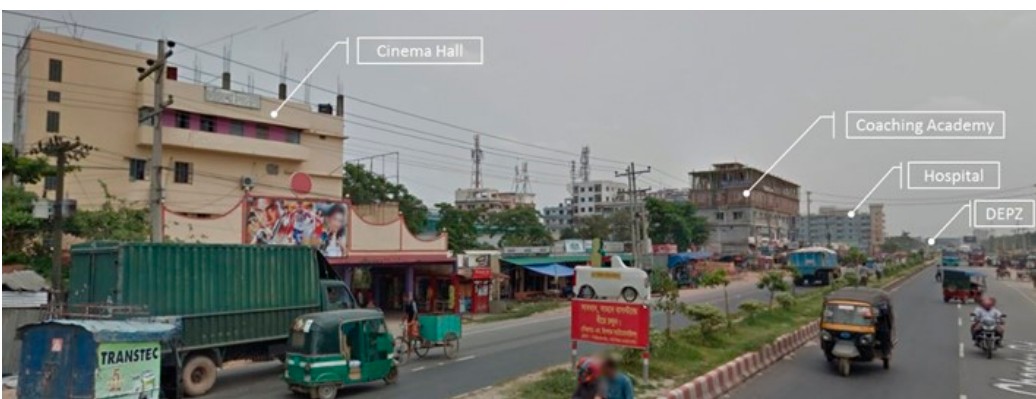

**Figure 12.** Google Street View image from May 2015 shows the roadside built-up area near DEPZ. Recreational facilities, health services, academic institutions, etc., are visible near DEPZ in addition to industrial and residential buildings.

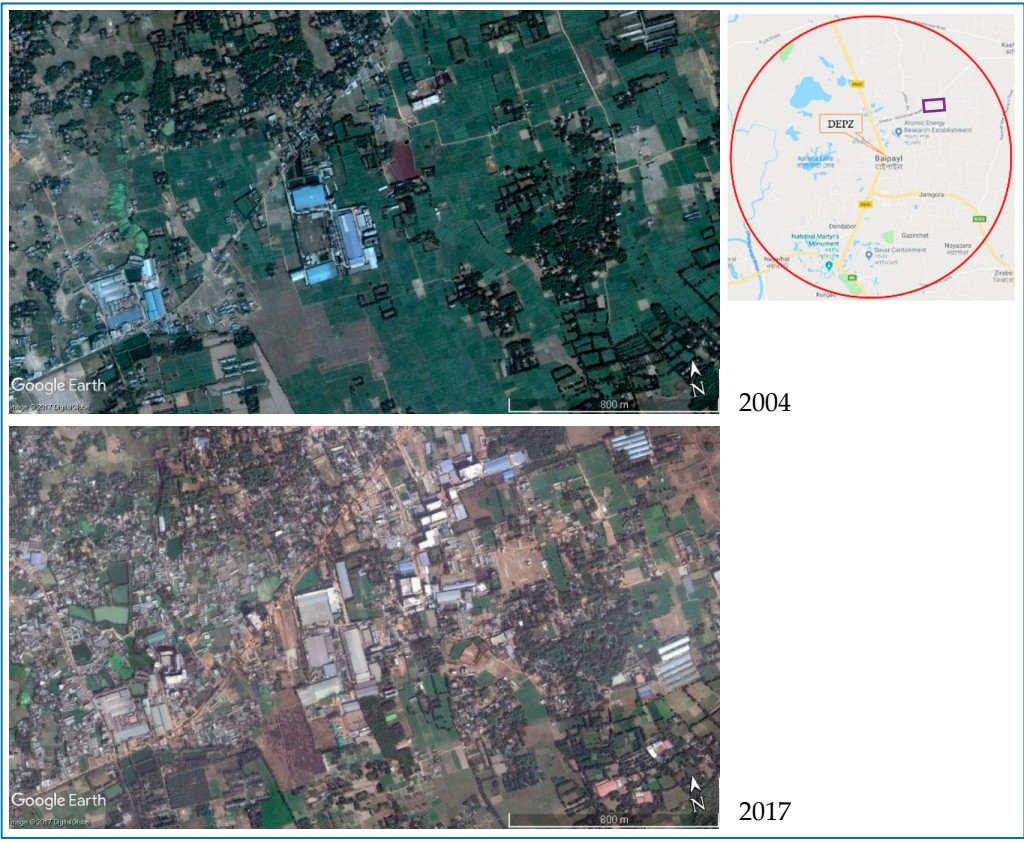

**Figure 13.** Google Earth images showing secondary industrial and urban growth between 2004 and 2017 in the north-east side of DEPZ.

The phenomena around DEPZ can be explained by Weber's theory of Location of Industries, which suggests that the location of the initial industry depends on variables such as the availability of raw materials, labor, and transportation facilities (Fales & Moses, 2017; Tiwari, 2014). Webber's theory, first proposed in early 1900, further states that subsequent industrial growth can happen because of the agglomeration effect, which utilizes the

existing infrastructure and availability of workforce and other resources (Wang & Ding, 2017). In other words, initial industries attract additional industries and people. The findings of this study suggest that DEPZ has worked as the nucleus for all subsequent industrial and residential growth in its surroundings.

In the first place, DEPZ was set up in its current location due to the proximity to the capital city, availability of flood-free land, and workforce [36]. To support DEPZ operation, new roads, and power and gas lines were constructed. Eventually, hundreds of secondary factories were established in the surroundings, taking advantage of the infrastructures, such as road network, power, and gas supply, which were initially developed for DEPZ [37]. People working in the factories, within and outside DEPZ, gradually started to live in the nearby areas; new residential units were constructed, and markets and other amenities were built [33].

This study further validates the claim of previous researchers [38–40] that the faster growth of urban built-up areas around DEPZ is associated with FDI-induced industrialization. In line with the findings of this study, the Dhaka Structure Plan (2016–2035) states that the DEPZ area has become a compact industrial agglomeration, and the accommodation and travel to and from the workplace were 'overlooked almost completely or were not very well planned and executed' [38]. As a result, the surrounding areas of the industrial zone have 'turned very quickly into unlivable high-density localities severely lacking in necessary infrastructure and services.' The report says the absence of proper urban local government bodies worsened the condition.

Multiple studies revealed that the FDI-induced industrialization in DEPZ and secondary unplanned industrial and urban growth are causing significant environmental degradation [41–44]. For instance, Mahbub et al. [41] identified substantial impacts because of effluent discharge from factories in DEPZ to surface water by analyzing parameters like pH, electrical conductivity, and total dissolved solids. Similarly, Mallick [42] indicated that untreated industrial waste from industries in DEPZ and surrounding areas was polluting rivers and other wetlands. Rahman et al. [45] found moderate-to-strong heavy metal contamination in agricultural soil near DEPZ in dry and wet seasons. Several other studies also concluded that the water quality of the DEPZ area has been deteriorating over time [41,46]. It could be noted here that DEPZ did not have any central effluent treatment plant (CETP) till 2012 [47,48]. Most DEPZ factories, therefore, have discharged untreated effluents in the nearby water bodies for about 20 years [44].

In addition to surface water degradation, the DEPZ area is also experiencing a gradual depletion of groundwater [49,50]. Industries inside and outside the DEPZ area, such as textile and dyeing [44], require large amounts of water to process their product [51–53]. These factories directly withdraw underground water as no surface water sources (rivers or other water bodies) are nearby. The increased population in the area also puts additional pressure on groundwater. Narratives from local people suggest that before establishing DEPZ, they would use ordinary hand pumps to withdraw drinking water. Now, they must go much deeper to extract water for domestic purposes, and standard hand pumps cannot reach the water table.

Bangladesh plans to set up many export-oriented economic zones, like the EPZs [54,55]. A new government body, Bangladesh Economic Zones Authority (BEZA), has been formed to establish economic zones in different areas of the country (http://www.beza.gov.bd/ (accessed on 14 May 2023)). However, before committing to those projects, it is advisable to look back at the long-term environmental and social impact of the existing industrial growth in the country. GIS and Remote Sensing methods can be employed to select the most suitable sites [56]. That will ensure sustainable urbanization and environmental protection to achieve sustainable development goals [57]. Bangladesh should also re-evaluate its environmental management and monitoring strategies because the existing environmental impact assessment (EIA) practice, approval, and enforcement processes failed to identify and mitigate the long-term adverse impacts of large-scale industrial development [58].

This study testifies that it is possible to classify older satellite images based on the known spectral responses of the latest images as the machine learning algorithm learns the spectral pattern from training or known samples. As the image classification model is based on GEE, it would be possible to re-run the model with minimum effort when newer images are captured. The study also shows that combining indices and machine learning techniques makes it possible to analyze older images for areas with a scarcity of reference data or ground truth information. With a simple modification, the classification model could be applied to industrial regions in Bangladesh or other parts of the world, even though GEE and machine learning algorithms are generally used for global- [59], continental- [16], national- [17], or regional-scale [60] investigations, these tools and techniques could be effectively applied in a local-level study like this one.

One of the limitations of this study is that it is difficult to separate the growth in urban built-up areas and population, which is directly related to the establishment of DEPZ. This is because DEPZ is in proximity to the country's capital city. One could argue that there would be some level of development by default, even if there were no EPZs in the area. To investigate that aspect, further study could be conducted by comparing the DEPZ area with another locality near Dhaka with similar proximity and settings. More studies could also be conducted to investigate the environmental and social degradation around DEPZ because of the accelerated urban growth. This study was conducted with freely available moderate resolution (30 m) Landsat satellite image. Future studies could be performed with higher-resolution satellite images, i.e., Spot, IKONOS, IRS, etc., or aerial photographs.

## 5. Conclusions

This study makes an original contribution to knowledge by investigating an industrial agglomeration's long-term environmental and social impact in a holistic manner. We have utilized 40 years' worth of satellite imagery as empirical evidence. In addition, Field surveys and key informant interviews were conducted to capture the strengths of the mixed-method approach.

Based on the analysis of satellite images, this study explores the association of FDI flow with urban growth, one of the critical sustainability indicators. This study used the DEPZ area as a case study, and freely available archived Landsat satellite images from GEE were used as empirical evidence. The result of image classification, based on a machine learning algorithm, shows that a significant portion of land around DEPZ has become a built-up area comprising secondary industries and residential development. It is now a settlement of over one million people. None of that development, however, was planned because the whole area is still managed under a local government structure that does not have a planning arm.

The area's population has also grown significantly more than the rest of the country. A positive association between the total built-up area and the population over time suggests that additional built-up areas attracted more people. Previous studies reported significant environmental degradation in the study area. Surface water in the area is becoming contaminated as most factories are discharging mostly untreated effluent. Subsurface water is depleting faster because most factories and settlements in the area are drawing water from the underground water table. Environmental degradation is linked to the growth of industrialization and built-up areas. Considering the combined effect of unplanned urban development and ecological consequences, it could be concluded that DEPZ, an FDI-induced industrial zone, has caused a situation of unsustainability in its proximity. The condition will be worsened if preventive measures are not taken.

GEE is an effective way to analyze historical satellite images with minimum effort and cost. The powerful image processing algorithms available in GEE can support complicated and resource-intensive analysis. There is a learning curve to code in GEE, but it is worth investing the time and effort to learn JavaScript to perform advanced research in the cloud platform. The study area was geographically concentrated in a small space, but analysis performed in GEE shows promising results. The limitation of this spatio-temporal

study could be avoided by analyzing commercially available higher-resolution images. Further studies are expected to explore the connection between FDI flow and environmental degradation with direct observational data.

**Supplementary Materials:** The shared Google Earth Engine JavaScript code used in this research can be found in the following link: https://code.earthengine.google.com/5b3a52fd47d3068ec1849ad53b7bdd58. This code could be used to reproduce the main analysis using GEE.

**Author Contributions:** Conceptualization, P.B., S.M. and T.F.G.; methodology, P.B., S.M., T.F.G. and S.D.; software, P.B.; validation, S.M., T.F.G. and S.D.; formal analysis, P.B.; investigation, P.B.; resources, P.B. and S.D.; data curation, P.B.; writing—original draft preparation, P.B.; writing—review and editing, P.B., S.M., T.F.G. and S.D.; visualization, P.B.; supervision, S.M. and T.F.G. All authors have read and agreed to the published version of the manuscript.

**Funding:** This research was supported by an Australian Government Research Training Program (RTP) Scholarship.

**Data Availability Statement:** All the data and materials supporting the results and analyses presented in this paper are available upon request.

**Acknowledgments:** This article is based on the Ph.D. thesis successfully submitted by the first author, Palash Basak, to the University of Newcastle, Australia, as below. It complies with the copywrite and publication regulations of the University of Newcastle, Australia. All maps, graphs, and tables are based on the following thesis. Palash, B. (2022). Foreign Direct Investment, Industrialisation, & Environmental Pollution in Bangladesh: An Analysis of Dhaka EPZ using Remote Sensing & GIS Techniques. University of Newcastle. http://hdl.handle.net/1959.13/1476712 (accessed on 15 May 2023).

**Conflicts of Interest:** The authors declare no conflict of interest.

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
