# Peer review of "Changing Land Use and Urban Dynamics around an Industrial Zone in Bangladesh: A Remote Sensing Analysis"

_land, doi:10.3390/land12091753_

Round 1

Reviewer 1 Report (Previous Reviewer 1)

This paper has been carefully revised in accordance with my review comments, although it does not fully address the issues I raised. On the whole, this paper can be self-consistent and logical. Moreover, in the conclusions part of the paper, the authors pointed out the shortcomings of this study and will use new data and methods to further study. More importantly, because of the typicality and practical significance of this study in Bangladesh, it is recommended to receive this paper after minor modifications.

Author Response

We have done another through review of the manuscript. Thank you for your time and effort to review our manuscript. Your comments have helped us improve the quality of our article. 

Reviewer 2 Report (Previous Reviewer 2)

In this manuscript, a change detection approach based on time-series Landsat images is presented. Topic is interesting but I think based on the following comments the manuscript is not ready for publication.

1-      Abstract: please mention steps of the method here. Moreover, results must be related the paper for example mention results obtained from landsat images.

2-      Introduction: style of this section is not appropriate. Please start with importance of this study. Then present literature review regarding the study. Literature review in this form is not strong. Finally, present research aims.

3-      Figures in the introduction is not usual. Please move them to the study area subsection.

4-      Research aims are not original, so I cannot recommend this manuscript to publish as an original research paper.

5-      There are some grammatical errors that should be tackled.

6-      Line 156: two different classifiers? Which one? I just saw SVM. Please mention some scientific descriptions regarding SVM in the text.

7-      Results of SVM method should be compared with other classifiers.

8-      Like introduction, I think the structure of the methodology section must be re-organized. For example, start with workflow and describe briefly steps of the method and then present subjects with more details.

9-      Validation regarding results is not presented in the text. So please present a confusion matrix regarding the accuracy of the classified maps.

-  Research aims are not original, so I cannot recommend this manuscript to publish as an original research paper.

Author Response

Reviewer 3 Report (New Reviewer)

The manuscript is well-written and matches the scope of this journal. In the overall scenario, the manuscript describes unplanned and haphazard urban growth. I recommended the following comments and suggestions to improve the quality of the manuscript. 

1. The title didn't match the scope of this manuscript. Change it according to the current study scope mentioned in the manuscript. Your main focus is urban-growth with industrialization and land planning. The sudden changes and association with water quality, quality, and agriculture are confusing. I will suggest replacing the agricultural land use with simple land use in the title.

2. The introduction section needs some rearrangements because your case study is very strong, but the case building is very poor. Rearrange the introduction section with storytelling and avoid unnecessary explanations. Strictly focused on the objectives of this study (new one). Also, follow the same pattern said in comment 1.

3. Figures 1, 2, and 3 show the same maps in different layouts. If necessary, add the source in Figure 3. 

4.  How do you find the association results (methods) is/are missing in the methodology section? If any, first elaborated in the methodology section

5. For accuracy assessment of the LULC maps, you mentioned 30 samples for one year image or LULC class?

6. Follow the same comments (general, 1 and 2) for the results and discussion sections

7. After making changes according to the comments and suggestions, add appropriate conclusive remarks in the conclusion section. 

8. Lastly, I recommended one article (read and cite) to examine land suitability and availability in a complex topographic environment. Especially to the last part will help you. https://doi.org/10.1016/j.apgeog.2021.102550ï¼› https://doi.org/10.1038/s41467-021-22702-2

The English language quality is quite well.

Round 2

Reviewer 2 Report (Previous Reviewer 2)

The authors revised the manuscript according to my comments. I think the type of manuscript is not original article. So, it is strongly recommended to publish it as a case study or other related type according to decision of the editor.

Author Response

We appreciate and respect the comment of the reviewer. However, we are unsure what the reviewer means by “original article.” As mentioned in the acknowledgment section of the manuscript, this article is based on the first author’s Ph.D. research (https://hdl.handle.net/1959.13/1476712), successfully submitted to the University of Newcastle, Australia. During the literature review, we have not encountered any research, especially in Bangladesh, where an industrial agglomeration's long-term environmental and social impact have been investigated holistically. We have utilized 40 years' worth of satellite imagery as empirical evidence. Field surveys and key informant interviews were conducted. Therefore, we think it is original research.

This manuscript is a resubmission of an earlier submission. The following is a list of the peer review reports and author responses from that submission.

Round 1

Reviewer 1 Report

1. This study is of great significance, revealing the changes of built-up area expansion and environmental pollution in the surrounding areas of Bangladesh's Dhaka Export Processing Zone (DEPZ) over the years, and also analyzing the possible causes of these changes, such as population growth. However, this paper fails to prove the causal relationship between population increase and built-up area growth.

2. The data and methods of regression analysis in this paper are not reliable, so the conclusion is not credible. For example, there are only 4 samples in the linear regression analysis of total population and built-up area, so the conclusion is very unreliable due to the small sample size. Moreover, this study fails to accurately calculate the population growth and built-up area expansion caused by DEPZ industrial zone. Therefore, it is recommended to supplement the data or use other appropriate methods.

3. For the above reasons, this paper is more like a research report, which is a development report on land expansion, population growth and environmental pollution caused by DEPZ industrial zone. However, an academic paper needs to raise the central question and prove it strictly, which is undoubtedly not done in this paper. Interestingly, the title of this paper is also more appropriate for a research report.

Reviewer 2 Report

In this paper, a change detection method based on classification is presented to detect buildup area from time-series images. Topic is interesting but the manuscript is not mature enough for publication. Please see some of comments below:

·         Title can be presented in a better form for example remote sensing and classification method can be mentioned.

·         Figure 1: please insert longitude and latitude in this figure.

·         Literature review must be presented in the introduction.

·         Line 107: descriptions regarding study area must be presented in subsection in the material and method section.

·         Please use a workflow to describe steps of the study.

·         Descriptions regarding classification must be presented in the text.

·         Validation regarding results i.e. classified maps should be done and presented in text.

·         Figure 5: please insert longitude and latitude in the plot.

·         Validation regarding change maps and change areas must be presented.